

# The beneficial effects of cumulus cells and oocyte-cumulus cell gap junctions depends on oocyte maturation and fertilization methods in mice

Cheng-Jie Zhou, Sha-Na Wu, Jiang-Peng Shen, Dong-Hui Wang, Xiang-Wei Kong, Angeleem Lu, Yan-Jiao Li, Hong-Xia Zhou, Yue-Fang Zhao and Cheng-Guang Liang

The Research Center for Laboratory Animal Science, College of Life Science, Inner Mongolia University, Hohhot, Inner Mongolia, China

## ABSTRACT

Cumulus cells are a group of closely associated granulosa cells that surround and nourish oocytes. Previous studies have shown that cumulus cells contribute to oocyte maturation and fertilization through gap junction communication. However, it is not known how this gap junction signaling affects *in vivo* versus *in vitro* maturation of oocytes, and their subsequent fertilization and embryonic development following insemination. Therefore, in our study, we performed mouse oocyte maturation and insemination using *in vivo-* or *in vitro*-matured oocyte-cumulus complexes (OCCs, which retain gap junctions between the cumulus cells and the oocytes), *in vitro*-matured, denuded oocytes co-cultured with cumulus cells (DCs, which lack gap junctions between the cumulus cells and the oocytes), and *in vitro*-matured, denuded oocytes without cumulus cells (DOs). Using these models, we were able to analyze the effects of gap junction signaling on oocyte maturation, fertilization, and early embryo development. We found that gap junctions were necessary for both *in vivo* and *in vitro* oocyte maturation. In addition, for oocytes matured *in vivo*, the presence of cumulus cells during insemination improved fertilization and blastocyst formation, and this improvement was strengthened by gap junctions. Moreover, for oocytes matured *in vitro*, the presence of cumulus cells during insemination improved fertilization, but not blastocyst formation, and this improvement was independent of gap junctions. Our results demonstrate, for the first time, that the beneficial effect of gap junction signaling from cumulus cells depends on oocyte maturation and fertilization methods.

## INTRODUCTION

Cumulus cells are defined as a group of closely associated granulosa cells that surround the oocyte and participate in the processes of oocyte maturation and fertilization. Cumulus cell function is dependent on gap junctions that form between cumulus cells and oocytes. Gap junctions are formed by aggregates of intercellular membrane channel proteins called connexins, which are members of a homologous family of more than 20

Corresponding author
Cheng-Guang Liang,
842505499@qq.com,
liangchengguang@gmail.com

proteins, and which allow for the exchange of small molecules between adjacent cells (*Kidder & Mhawi, 2002*).

Fully grown oocytes become competent to undergo two aspects of maturation: cytoplasmic and nuclear. Both of these processes are essential for the formation of an oocyte with the capacity to undergo fertilization and development to live offspring. Nuclear maturation encompasses the processes of reversing meiotic arrest at prophase I and driving the progression of meiosis to metaphase II. Cytoplasmic maturation refers to the processes that prepare the egg for activation and preimplantation development (*Eppig, 1996*).

Gap junctions between cumulus cells and oocytes are thought to be essential for oocyte maturation and fertilization. During oocyte nuclear maturation, gap junctions are the main connection between cumulus cells and oocytes (*Feng et al., 2013*; *Furger et al., 1996*; *Santiquet et al., 2012*), and they allow a rapid transfer of small metabolites and regulatory molecules from the cumulus cells into the oocyte (*Van Soom et al., 2002*). During *in vitro* maturation (IVM), the gap junction protein Cx43 plays a functional role in gap junction breakdown in the oocyte-cumulus complexes (OCCs) (*Sasseville et al., 2009*). During fertilization, the cumulus cells attract (*Eisenbach, 1999*), trap (*Bedford & Kim, 1993*), and select spermatozoa (*Carrell et al., 1993*), and prevent the premature hardening of the zona pellucida (ZP) (*Tanghe et al., 2002*), all of which are necessary for successful fertilization. Furthermore, gap junctional communication between oocytes and cumulus cells has been shown to be an essential factor in supporting fertilization (*Tanghe et al., 2003*). To some extent, early embryo development depends on the successful coordination of processes that occur during oocyte cytoplasmic maturation, including molecular changes, organelle reorganization, and cytoskeletal changes (*Damiani et al., 1996*; *Reyes & Ross, 2016*; *Salamone et al., 2001*). Numerous studies have shown that the presence of cumulus cells can improve cytoplasmic maturation (*Ikeda & Yamada, 2014*; *Tanghe et al., 2002*). However, it is still not clear if cumulus cells and their gap junctions with oocytes are necessary for the successful *in vivo* and *in vitro* maturation of oocytes. Furthermore, there is currently a lack of understanding of the contribution of cumulus cells and gap junctions to fertilization and early embryo development downstream of oocyte maturation.

Thus, in this study, we used four types of oocytes for our maturation and insemination procedures, including *in vivo*- or *in vitro*-matured oocyte-cumulus complexes (OCCs, which possess cumulus cells and intact gap junctions), *in vitro*-matured, denuded oocytes co-cultured with cumulus cells (DCs, which have cumulus cells, but lack gap junctions), and *in vitro*-matured, denuded oocytes without cumulus cells (DOs). Using these culture models, we investigated oocyte maturation, fertilization, and early embryo development to evaluate contributions of cumulus cells and oocyte-cumulus cell gap junctions to each of these processes.

## MATERIALS AND METHODS

### Ethics statement and animal feeding regimens

All studies adhered to procedures consistent with the National Research Council Guide for the Care and Use of Laboratory Animals and were approved by the Institutional Animal

Care and Use Committee at the Inner Mongolia University (Approval number: SYXK 2014-0002). Mice were maintained under the care of the Laboratory Animal Facility at Inner Mongolia University. Mice were kept at a constant temperature of $22 \pm 2$ °C on a 12 h light/dark cycle and had unrestricted access to food and water.

## Oocyte collection
Adult female (B6D2) F1 mice (4–8 weeks of age) were used for the oocyte collections. All chemicals and media were purchased from Sigma-Aldrich Company (St. Louis, MO, USA) unless stated otherwise. The germinal vesicle (GV) stage oocytes were collected by puncturing the follicles of ovaries at 48 h after pregnant mare serum gonadotropin (PMSG; SanSheng, Ningbo, China) injection. The cumulus cells were removed by gentle pipetting. For *in vivo* metaphase II (MII) stage oocyte collection, mice were superovulated by injection of 10 IU PMSG, followed by injection of 10 IU human chorionic gonadotropin (hCG; SanSheng, Ningbo, China) 48 h later. The cumulus cells were dispersed by 0.3 mg/mL hyaluronidase in HEPES-M2 medium.

## Oocyte maturation
GV oocytes were cultured in MEM Alpha ($\alpha$-MEM; Gibco, Pleasanton, CA, USA) medium supplemented with 5% (v/v) fetal calf serum (FCS), 10 ng/mL epidermal growth factor (EGF), and 0.01 AU/mL follicle-stimulating hormone (FSH), under a humidified atmosphere of 5% $CO_2$ at 37 °C for 14–16 h. The number of oocytes with the first polar body (PB1) was counted to determine the percentage of nuclear maturation.

## IVF and embryo culture
Adult male (B6D2) F1 mice (12–14 weeks of age) were used for the sperm collections. The sperm suspension was capacitated for 2 h in 200 μL T6 medium supplemented with 10 mg/mL bovine serum albumin (BSA). MII oocytes were incubated with spermatozoa for 6 h in 200 μL T6 medium supplemented with 20 mg/mL BSA. The sperm concentration used for fertilization was $1 \times 10^6$/mL. The zygotes were collected and cultured in Chatot-Ziomet-Bavister (CZB) medium containing 3 mg/ml BSA without glucose under a humidified atmosphere of 5% $CO_2$ at 37 °C for the first 2 days, and then transferred to CZB medium supplemented with 5.5 mmol/L glucose when the embryos reached the 4-cell stage. The percentage of embryos that reached the 2-cell stage was used for fertilization evaluation. The embryos were checked at 48, 72, and 96 h after fertilization to calculate the percentage of 4-cell, morula, and blastocyst stages, respectively.

## Experimental design
In our study, oocytes were randomly divided into four groups to mature: (1) OCCs from *in vivo* maturation, which we abbreviated as M-vivo-OCC; (2) GV oocytes with at least three layers of attached cumulus cells for IVM, which we abbreviated as M-vitro-OCC; (3) GV-stage denuded oocytes co-cultured with dispersed cumulus cells (DC) for IVM, which we abbreviated as M-vitro-DC; (4) GV-stage DOs for IVM alone, which we abbreviated as M-vitro-DO. For subsequent insemination, we co-incubated sperm with OCC (I-OCC), DC (I-DC), or DO (I-DO). For cumulus cell and oocyte co-culture, cumulus cells were

collected from GV-stage OCCs by pipetting. The cumulus cells collected from each OCC were supplemented back to the corresponding oocyte. The combinations of different maturation procedures and different insemination procedures were tested to evaluate the roles of cumulus, resulting in a total of ten combinations (M-vivo-OCC + I-OCC, M-vivo-OCC + I-DC, M-vivo-OCC + I-DO, M-vitro-OCC + I-OCC, M-vitro-OCC + I-DC, M-vitro-OCC + I-DO, M-vitro-DC + I-DC, M-vitro-DC + I-DO, M-vitro-DO + I-DC, and M-vitro-DO + I-DO), which are shown in Fig. 1.

## Statistical analysis

The data are presented as the means $\pm$ standard deviation (SD) from three replicate experiments. Differences were evaluated using the Student's $t$ test. $P < 0.05$ was regarded as statistically significant.

## RESULTS

### The role of cumulus cells and their gap junctions in oocyte maturation

To investigate the contribution of cumulus cells and their gap junctions during oocyte maturation, PB1 extrusion was calculated using the following maturation models, as shown in Fig. 1: M-vivo-OCC, M-vitro-OCC, M-vitro-DC, and M-vitro-DO. Our results showed that the highest nuclear maturation percentage was obtained among the OCC groups, but there was no statistical difference between the *in vivo* and *in vitro* OCC models. However, if cumulus cells were removed from the oocytes, PB1 extrusion was significantly decreased ($P < 0.01$), and this decrease was not reversed by co-culturing cumulus cells with oocytes ($P < 0.01$) (Fig. 2 and Table S1), suggesting that intact gap junctions between the cumulus cells and oocytes are necessary for efficient oocyte maturation.

### The role of cumulus cells and their gap junctions in oocyte fertilization

Fertilization can be affected by both oocyte maturation and insemination procedures. Thus, to assess the effects of cumulus cells and gap junctions during maturation and insemination, we normalized the fertilization percentage by quantifying the number of 2-cell embryos relative to the number of total GV-stage oocytes or total MII-stage oocytes in each group.

#### *Fertilization based on the number of GV oocytes*

Using the same maturation method, we observed significant increases in the percentage of 2-cell-stage embryos when comparing oocytes cultured with, to those cultured without, cumulus cells during insemination ($P < 0.01$) (Fig. 3A and Table S2), suggesting that the presence of cumulus cells improves fertilization. For the oocytes matured *in vivo* (M-vivo-OCC), detachment of the cumulus cells from the oocytes before insemination, which disrupted the gap junctions between the cells, reduced the fertilization percentage ($P < 0.01$). However, under the same conditions, fertilization was not affected for the OCCs matured *in vitro* (M-vitro-OCC) (Fig. 3A and Table S2).

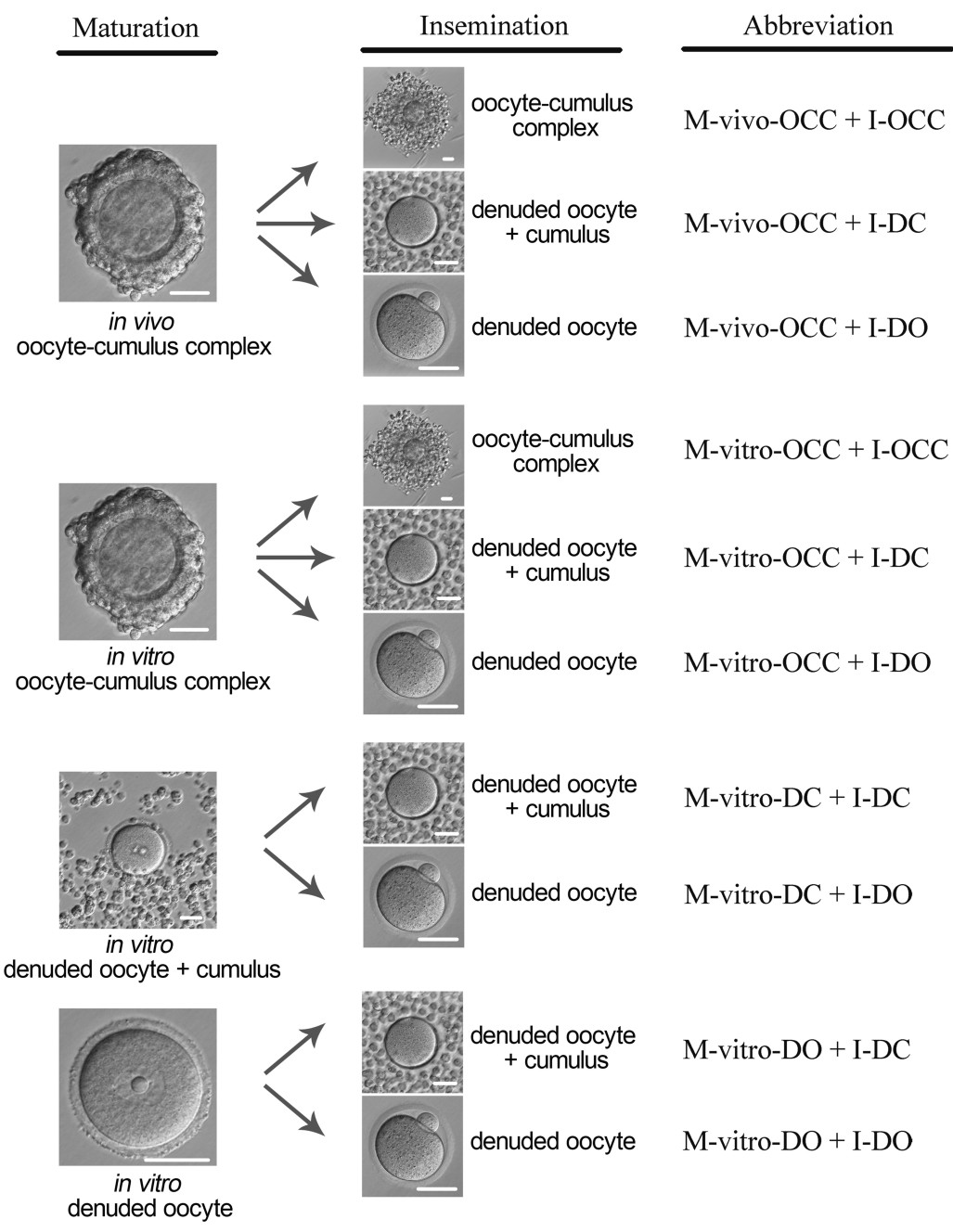

**Figure 1 Experimental design.** *In vivo* MII stage oocyte-cumulus complexes (OCC) were collected from oviduct in mice superovulated by PMSG, followed by hCG. GV stage OCCs were collected from ovaries of the mice 48 h after the administration of PMSG. Cumulus cells were removed by gentle pipetting. Oocytes were divided into four groups for maturation: *in vivo* OCC, *in vitro* OCC, *in vitro* denuded oocyte (DO) + cumulus cells, and *in vitro* DO. For OCCs matured *in vivo* or *in vitro*, oocytes were divided into three groups for insemination: OCC, DO + cumulus, and DO. For *in vitro* matured DOs with or without cumulus cells, oocytes were divided into two groups for insemination: DO + cumulus and DO. The abbreviations used are listed at far right of each maturation and insemination model. Scale = 50 μm.

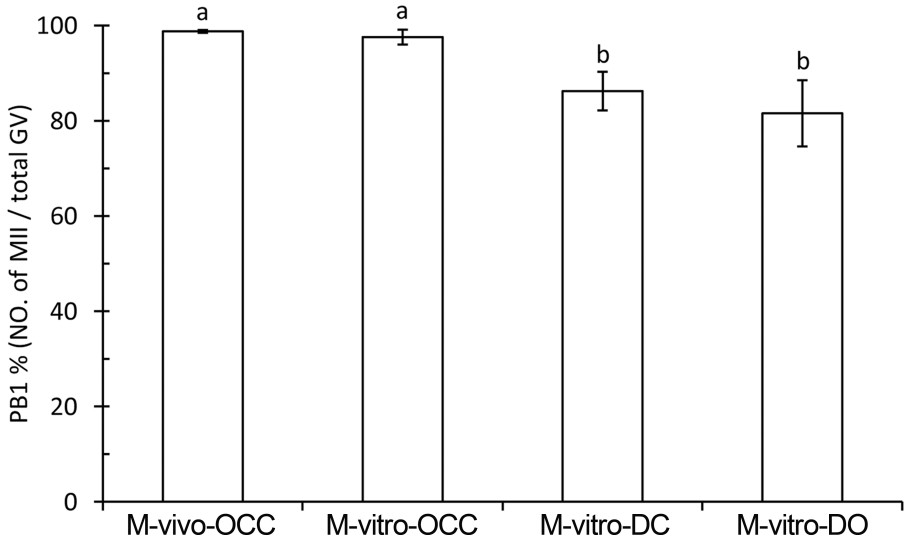

**Figure 2** **Percent of PB1 under different maturation methods.** Oocytes were divided into four groups for maturation: M-vivo-OCC, M-vitro-OCC, M-vitro-DC, and M-vitro-DO. The percentages of the PB1 were calculated. Percentages without a common letter are statistical significant different ($P < 0.05$).

Next, we analyzed the data to investigate the influence of maturation methods on fertilization outcomes. Using the same insemination method, successful fertilization was dependent on the maturation method of the oocytes. For example, the oocytes matured *in vivo* (M-vivo-OCC) showed the highest fertilization percentage, the OCCs matured *in vitro* (M-vitro-OCC) showed a median fertilization percentage, and the oocytes matured *in vitro* with dispersed cumulus cells (M-vitro-DC) or without cumulus cells (M-vitro-DO) showed the lowest fertilization percentages ($P < 0.01$) (Fig. 3A and Table S2).

### Fertilization based on the number of MII oocytes

To evaluate the contribution of cumulus cells and gap junctions during insemination, we quantified the percentage of 2-cell-stage embryos among the total MII oocytes. Using the same maturation method, we found that the presence of cumulus cells in the insemination medium improved the fertilization percentages ($P < 0.01$) (Fig. 3B and Table S3). For the M-vivo-OCC oocytes, the removal of cumulus cells from oocytes during insemination (I-DO) reduced the fertilization percentages; however, this reduction could be partially reversed by the addition of cumulus cells into the insemination medium (I-DC) ($P < 0.01$) (Fig. 3B and Table S3). Using the same insemination method, the oocytes matured *in vivo* had higher fertilization percentages than those matured *in vitro* ($P < 0.01$), but the presence of cumulus cells during IVM did not affect fertilization ($P > 0.05$) (Fig. 3B and Table S3).

### The role of cumulus cells and their gap junctions in embryo development

To investigate the contributions of cumulus cells to early embryo development, we calculated the percentages of 4-cell, morula, and blastocyst stages following insemination. Successful embryo development is determined by three important factors: the quality of MII

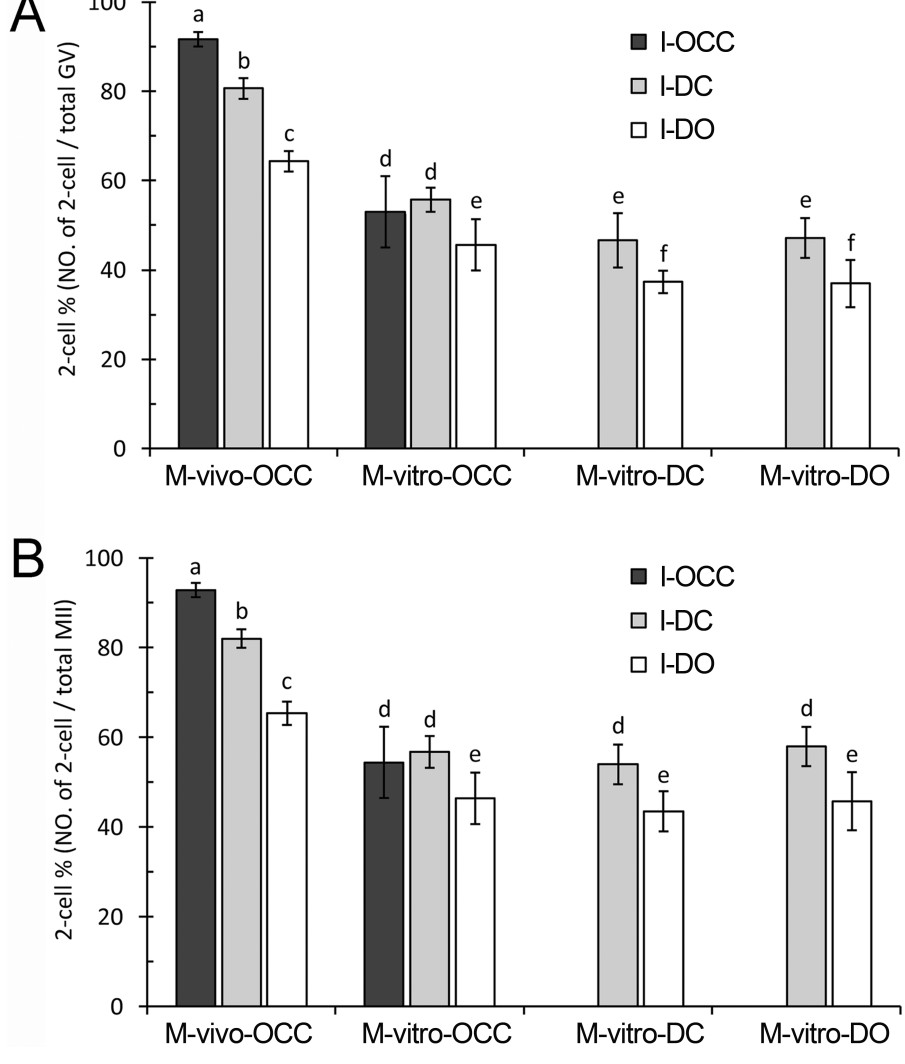

**Figure 3 Fertilization of oocytes under different maturation and insemination combinations.** (A) Percent of fertilization based on the number of GV oocytes. (B) Percent of fertilization based on the number of MII oocytes. Percentages without a common letter are statistical significant different ($P < 0.05$).

oocytes (i.e., the progression from GV stage to MII stage), the procedure of insemination (i.e., the progression from MII stage to 2-cell stage), and the development starting from the 2-cell stage (i.e., the progression after 2-cell stage). To assess the effects of cumulus cells and gap junctions during maturation, insemination, and early embryo development, we quantified each embryonic stage as a percentage of the total number of GV-stage oocytes, MII-stage oocytes, and 2-cell-stage embryos, respectively.

### Embryonic development based on the number of GV oocytes

For the oocytes matured *in vivo* (M-vivo-OCC), we observed a significant increase in the percentage of embryos in groups cultured with cumulus cells during insemination, compared with those without cumulus cells. More specifically, the M-vivo-OCC oocytes that retained gap junctions with cumulus cells during insemination (I-OCC) had the highest

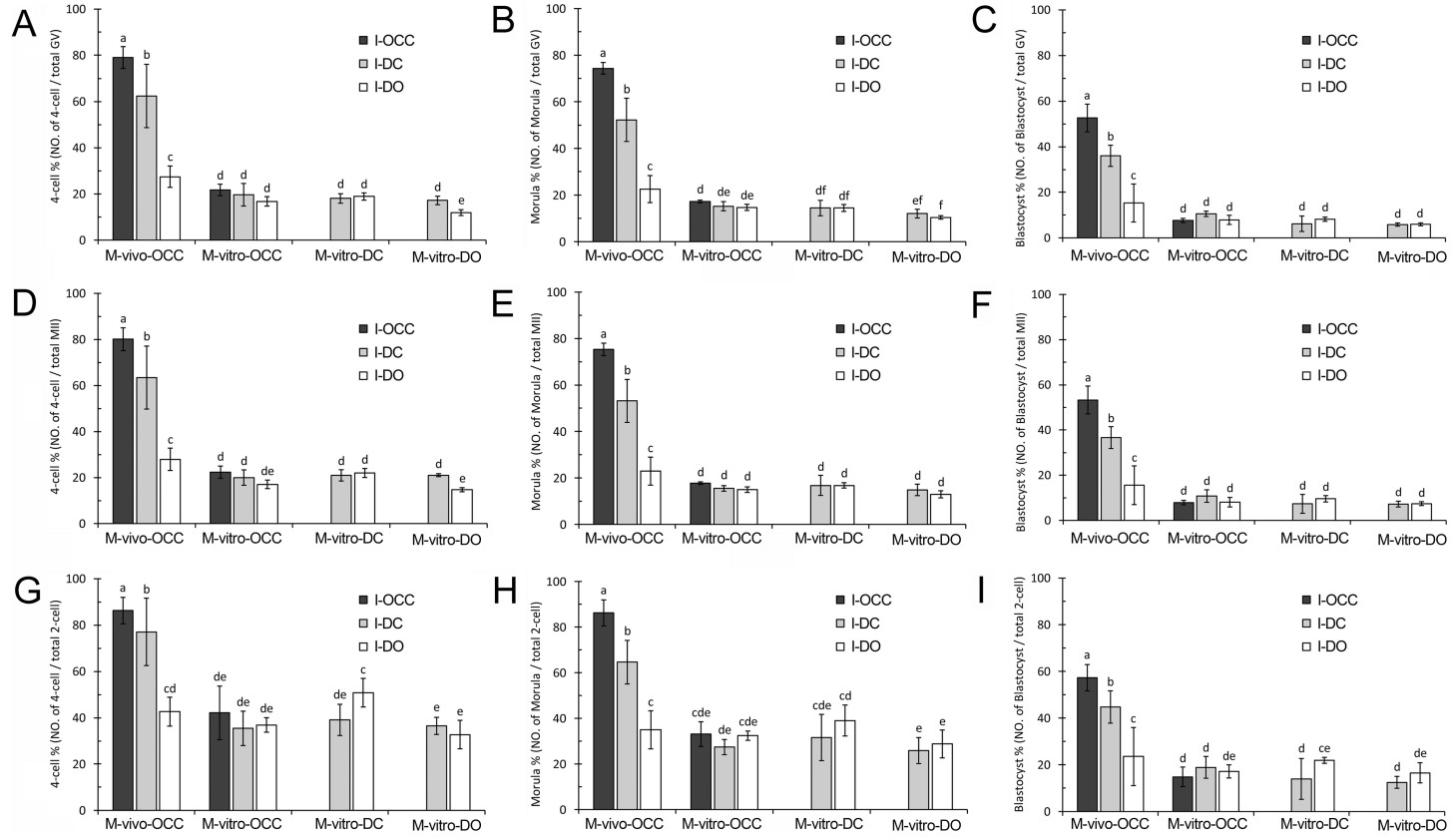

**Figure 4 Development of embryos generated from combinations of different maturation and insemination methods.** (A–C), Embryonic development based on the number of GV oocytes. (D–F), Embryonic development based on the number of MII oocytes. (G–I), Embryonic development based on number of 2-cell embryos. Percentages without a common letter are statistical significant different ($P < 0.05$).

embryo development percentage, those inseminated in the presence of cumulus cells but without gap junctions (I-DC) showed a median embryo development percentage, and those inseminated in the absence of cumulus cells (I-DO) had the lowest embryo development percentage (4-cell stage: $P < 0.01$, Fig. 4A and Table S4; morula stage: $P < 0.01$, Fig. 4B and Table S4; and blastocyst stage: $P < 0.01$, Fig. 4C and Table S4).

For the oocytes matured *in vitro* with cumulus cells (M-vitro-OCC and M-vitro-DC), the loss of oocyte-cumulus cell gap junctions (I-DC) or the absence of cumulus cells (I-DO) during insemination did not affect early embryo development (4-cell stage: $P > 0.05$, Fig. 4A and Table S4; morula stage: $P > 0.05$, Fig. 4B and Table S4; and blastocyst stage: $P > 0.05$, Fig. 4C and Table S4).

Interestingly, for the oocytes matured *in vitro* without cumulus cells (M-vitro-DO), the absence of cumulus cells during insemination (I-DO) reduced the percentage of 4-cell embryos ($P < 0.05$) (Fig. 4A and Table S4). However, this absence did not affect formation of the morula ($P > 0.05$) (Fig. 4B and Table S4) or blastocyst stages ($P > 0.05$) (Fig. 4C and Table S4).

For all types of oocytes matured *in vitro* (M-vitro-OCC, M-vitro-DC, and M-vitro-DO), the presence of cumulus cells during IVM or insemination did not alter blastocyst formation ($P > 0.05$) (Fig. 4C and Table S4).

Notably, the oocytes matured *in vivo* (M-vivo-OCC), regardless of the presence or absence of cumulus cells during insemination (I-OCC, I-DC, and I-DO), had higher embryo development percentages than those matured *in vitro* (M-vitro-OCC, M-vitro-DC, and M-vitro-DO) (4-cell stage: $P < 0.01$, Fig. 4A and Table S4; morula stage: $P < 0.01$, Fig. 4B and Table S4; and blastocyst stage: $P < 0.01$, Fig. 4C and Table S4).

### Embryonic development based on the number of MII oocytes

To evaluate the contribution of cumulus cells and gap junctions during insemination, we quantified the percentage of embryos from each stage relative to the total number of MII oocytes. For the *in vivo*-matured oocytes (M-vivo-OCC), we observed a greater percentage of embryo development when cumulus cells were present during insemination. More specifically, if cumulus cells retained gap junctions with oocytes (I-OCC), the highest embryo development percentage was obtained. If cumulus cells were separated from oocytes, but were still present in the insemination medium (I-DC), a median embryo development percentage was obtained. If cumulus cells were removed from the insemination medium (I-DO), the percentage of embryo development was much lower (4-cell stage: $P < 0.01$, Fig. 4D and Table S5; morula stage: $P < 0.01$, Fig. 4E and Table S5; and blastocyst stage: $P < 0.01$, Fig. 4F and Table S5).

For the oocytes matured *in vitro* with cumulus cells (M-vitro-OCC and M-vitro-DC), the absence of cumulus cells during insemination (I-DO) did not affect early embryo development (4-cell stage: $P > 0.05$, Fig. 4D and Table S5; morula stage: $P > 0.05$, Fig. 4E and Table S5; and blastocyst stage: $P > 0.05$, Fig. 4F and Table S5).

For the oocytes matured *in vitro* without cumulus cells (M-vitro-DO), the absence of cumulus cells during insemination (I-DO) reduced the percentage of 4-cell embryos ($P < 0.05$) (Fig. 4D and Table S5), but did not affect the percentages of morula or blastocyst embryos (morula stage: $P > 0.05$, Fig. 4E and Table S5; blastocyst stage: $P > 0.05$, Fig. 4F and Table S5).

For the oocytes matured *in vitro* (M-vitro-OCC, M-vitro-DC, and M-vitro-DO), the absence of cumulus cells during IVM and insemination (I-DO) did not affect blastocyst formation ($P > 0.05$) (Fig. 4F and Table S5).

The oocytes matured *in vivo* (M-vivo-OCC), regardless of the presence or absence of cumulus cells during insemination (I-OCC, I-DC, and I-DO), had a greater percentage of embryo development than those matured *in vitro* (M-vitro-OCC, M-vitro-DC and M-vitro-DO) (4-cell stage: $P < 0.01$, Fig. 4D and Table S5; morula stage: $P < 0.01$, Fig. 4E and Table S5; and blastocyst stage: $P < 0.01$, Fig. 4F and Table S5).

### Embryonic development based on number of 2-cell embryos

To evaluate the contribution of cumulus cells and gap junctions after insemination, we quantified the percentage of embryos from each stage relative to the total number of 2-cell embryos. For the oocytes matured *in vivo* (M-vivo-OCC), a high percentage of embryo development was dependent on the presence of cumulus cells and their gap junctions with oocytes during insemination. For the M-vivo-OCC oocytes that retained gap junctions with cumulus cells during insemination (I-OCC), we observed the highest embryo development percentage. If cumulus cells were separated from

oocytes, but were still present in the insemination medium (I-DC), a median embryo development percentage was obtained. If cumulus cells were removed from the oocytes during insemination (I-DO), the lowest embryo development percentage was obtained (4-cell stage: $P < 0.01$, Fig. 4G and Table S6; morula stage: $P < 0.01$, Fig. 4H and Table S6; and blastocyst stage: $P < 0.01$, Fig. 4I and Table S6).

For the OCCs matured *in vitro* (M-vitro-OCC), the absence of cumulus cells during insemination (I-DO) did not affect embryo development (4-cell stage: $P > 0.05$, Fig. 4G and Table S6; morula stage: $P > 0.05$, Fig. 4H and Table S6; and blastocyst stage: $P > 0.05$, Fig. 4I and Table S6). Similar results were obtained with DOs matured *in vitro* (M-vitro-DO) (4-cell stage: $P > 0.05$, Fig. 4G and Table S6; morula: $P > 0.05$, Fig. 4H and Table S6; and blastocyst: $P > 0.05$, Fig. 4I and Table S6).

However, for the oocytes matured *in vitro* with cumulus cells (M-vitro-DC), the absence of cumulus cells during insemination increased the percentage of 4-cell-stage and blastocyst, but not morula embryos (4-cell stage: $P < 0.05$, Fig. 4G and Table S6; and morula stage: Fig. 4H and Table S6; and blastocyst stage: $P < 0.05$, Fig. 4I and Table S6).

Compared to the oocytes matured *in vitro* (M-vitro-OCC, M-vitro-DC, and M-vitro-DO), the presence of cumulus cells during insemination (I-OCC and I-DC) increased the percentage of embryo development from *in vivo*-matured oocytes (M-vivo-OCC) (4-cell stage: $P < 0.01$, Fig. 4G and Table S6; morula stage: $P < 0.01$, Fig. 4H and Table S6; and blastocyst stage: $P < 0.01$, Fig. 4I and Table S6).

## DISCUSSION

### Cumulus cells and oocyte-cumulus cell gap junctions contribute to oocyte maturation

In this study, the contribution of cumulus cells and oocyte-cumulus cell gap junctions to oocyte maturation was analyzed under different culture conditions, including *in vivo* or *in vitro* maturation, with or without cumulus cells or gap junctions. This comprehensive experimental design allowed us to better understand the functions of cumulus cells and oocyte-cumulus cell gap junctions during maturation. Previous studies have shown that gap junctions between cumulus cells and oocytes are the main channels allowing the exchange of ions and small molecules, such as purines (*Downs & Eppig, 1986*; *Eppig, Ward-Bailey & Coleman, 1985*) and cyclic adenosine monophosphate (cAMP) (*Kumar & Gilula, 1996*; *Yoshimura et al., 1992*), to inhibit resumption of premature oocyte meiotic progression. Our study demonstrates that these gap junctions are also essential for oocytes to achieve higher nuclear maturation rates. We showed that disruption of these junctions impaired PB1 extrusion, and that this impairment could not be rescued by culturing dispersed cumulus cells with oocytes, which is consistent with previous studies showing that co-culturing with dispersed cumulus cells does not improve the nuclear maturation of oocytes (*Downs, 2001*; *Luciano et al., 2005*; *Tao et al., 2008*).

In addition, in our study, we used maturation medium containing EGF and FSH. A previous study in rabbit showed that the presence of growth factors in culture systems facilitates nuclear maturation in OCCs but not in DOs (*Lorenzo et al., 1996*), which

is similar to what we observed. For oocyte growth and development *in vitro*, the gap junctions between cumulus cells and oocytes must be maintained (*Eppig, 1979*; *Hashimoto et al., 1998*; *Tanghe et al., 2002*). Once gap junctions are disrupted by removing the cumulus cells, the passage of necessary signaling molecules is interrupted. Consistent with previous data, we demonstrated that gap junction signaling between oocytes and cumulus cells was critical for oocyte maturation.

## Cumulus cells and oocyte-cumulus cell gap junctions contribute to oocyte fertilization

Although cumulus cell function during oocyte maturation has been widely studied, less is known about cumulus cell contribution during insemination. To the best of our knowledge, this study is the first to demonstrate that cumulus cells affected fertilization by influencing *in vitro* oocyte nuclear maturation. Using the same insemination method, successful fertilization depended on the method of oocyte nuclear maturation, with *in vivo*-matured oocytes showing the highest fertilization percentages. Using the same IVM method, the presence of cumulus cells during insemination resulted in higher cleavage rates. Therefore, our data suggest that *in vivo* OCC maturation followed by OCC insemination is the optimal combination for obtaining the highest fertilization percentage.

A new finding in our study was that when *in vitro*-matured MII oocytes were used for fertilization, the presence of cumulus cells during insemination improved the cleavage percentages, even when the gap junctions were destroyed. It has been proposed that gap junctional communication between the oocyte and corona cells is needed for supporting fertilization (*Tanghe et al., 2003*). We hypothesize that the improved fertilization percentage obtained when the DOs were inseminated in the presence of cumulus cells, but in the absence of gap junctions, was due to secreted factors from the cumulus cells (*Guidobaldi et al., 2008*; *Sun et al., 2005*). This hypothesis is quite plausible, as cumulus cells have been shown to secrete chemotactic factors that guide the spermatozoon to the oocyte (*Ito, Smith & Yanagimachi, 1991*; *Sun et al., 2005*), which increases the chance of fertilization.

Another new finding of our study was that *in vivo*-matured oocytes achieved higher cleavage percentages than those from IVM, even in the absence of cumulus cells during insemination. To explain these data, we hypothesize that the *in vivo*-matured oocytes had undergone more complete cytoplasmic maturation than the *in vitro*-matured oocytes. Moreover, for the oocytes matured *in vivo*, removal of cumulus cells during insemination (I-DO) reduced the fertilization percentage, but this reduction was partially reversed by the addition of cumulus cells to the insemination medium. Interestingly, for the *in vitro*-matured OCCs, this reduction was reversed completely by adding cumulus cells to the insemination medium. These data suggest that the gap junctions of the *in vitro*-matured OCCs are defective, leading to the reduced fertilization percentages during IVF compared with the *in vivo*-matured OCCs.

## Cumulus cells and oocyte-cumulus cell gap junctions contribute to early embryo development

There is an existing debate concerning whether or not cumulus cells contribute to oocyte cytoplasmic maturation, a process that is necessary for early embryo development. It has

been suggested that co-culturing DOs embedded in cumulus cell clumps can improve cytoplasmic maturation (*Feng et al., 2013*), and denuding mouse oocytes of cumulus cells impairs *in vitro* cytoplasmic maturation (*Ge et al., 2008*). However, our data support the concept that cumulus cells do not contribute to cytoplasmic maturation during IVM. These different outcomes may be due to the employment of different species, different standards for evaluating cytoplasmic maturation, or different maturation conditions used in each study.

Our study showed that oocytes matured *in vivo* were the most suitable for embryo development, indicating that *in vivo*-matured oocytes possess more complete cytoplasmic maturation than *in vitro*-matured oocytes. Previously, pronuclear (PN) transfer between *in vivo*-matured and *in vitro*-matured oocytes was used to examine the nuclear-ooplasm effects on resultant embryo development. The results showed that as long as the ooplasm was derived from *in vivo* samples, reconstructed embryos had a higher developmental ability regardless of PN origin (*Chang et al., 2005*). These data showed that the ooplasm is the decisive factor for embryo development, which is consistent with our hypothesis.

In conclusion, our study systematically evaluated the contribution of cumulus cells and oocyte-cumulus cell gap junctions to oocyte maturation, fertilization, and embryo development, using various oocyte maturation and insemination methods. Our results demonstrate that the beneficial effects of cumulus cells and oocyte-cumulus cell gap junctions depend on oocyte maturation and insemination methods.

### Funding
This work was supported by the National Natural Science Foundation of China (31160243, 31371454), the Natural Science Foundation of Inner Mongolia Autonomous Region of China (2015JQ02), Program of Science and Technology Supporting for Homecoming People in Inner Mongolia to CL and Program of Graduate Student Education and Innovation Plan in Inner Mongolia (B20151012615) to CZ. The funders had no role in study design, data collection and analysis, decision to publish, or preparation of the manuscript.

### Grant Disclosures
The following grant information was disclosed by the authors:
National Natural Science Foundation of China: 31160243, 31371454.
Natural Science Foundation of Inner Mongolia Autonomous Region of China: 2015JQ02.
Program of Science and Technology.
Program of Graduate Student Education and Innovation Plan in Inner Mongolia: B20151012615.

### Competing Interests
The authors declare there are no competing interests.

## Author Contributions

- Cheng-Jie Zhou conceived and designed the experiments, performed the experiments, analyzed the data, contributed reagents/materials/analysis tools, wrote the paper, prepared figures and/or tables, reviewed drafts of the paper.
- Sha-Na Wu and Jiang-Peng Shen performed the experiments, contributed reagents/materials/analysis tools.
- Dong-Hui Wang, Xiang-Wei Kong, Angeleem Lu, Yan-Jiao Li and Yue-Fang Zhao performed the experiments.
- Hong-Xia Zhou conceived and designed the experiments, performed the experiments.
- Cheng-Guang Liang conceived and designed the experiments, analyzed the data, wrote the paper, prepared figures and/or tables, reviewed drafts of the paper.

## Animal Ethics

The following information was supplied relating to ethical approvals (i.e., approving body and any reference numbers):

All studies adhered to procedures consistent with the National Research Council Guide for the Care and Use of Laboratory Animals and were approved by the Institutional Animal Care and Use Committee at the Inner Mongolia University. Mice were maintained under the care of the Laboratory Animal Facility, Inner Mongolia University.

The approval number is SYXK 2014-0002.

## Data Availability

The raw data was supplied as Data S1.

## Supplemental Information

Supplemental information for this article can be found online at http://dx.doi.org/10.7717/peerj.1761#supplemental-information.

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
