# Peer review of "The beneficial effects of cumulus cells and oocyte-cumulus cell gap junctions depends on oocyte maturation and fertilization methods in mice"

_PeerJ, doi:10.7717/peerj.1761_

## Round 0.1 · original submission · Major Revisions

All reviewers had positive comments, however, point to point responses and a major revision is required.

Reviewer 1 ·

Basic reporting

No Comments

Experimental design

No Comments

Validity of the findings

No Comments

Additional comments

Cumulus cells have multiple functions during oocyte maturation and insemination. The authors investigated the function of the cumulus systematically by four models: In vivo oocyte-cumulus complexes (OCC), in vitro OCC, in vitro denuded oocytes (DO) + cumulus cells and in vitro DO. Their results showed that cumulus cells play key roles during oocytes maturation and insemination. Cumulus cell contribution is dependent on the connections of their gap junction with oocytes. This is a valuable topic. In general, the methods used in this study are feasible and the data are solid to support the conclusion. It could be taken into consideration for publication

·

Basic reporting

No Comments

Experimental design

No Comments

Validity of the findings

No Comments

Additional comments

Coordination between cumulus cells and oocytes within the ovary is crucial for follicle development and oocyte maturation. Cumulus-oocyte co-culture in vitro is a good model to study this complicated coordinative mechanism. Many related researches have been done. However, the most prominent superiority of the present study is that it not only studied oocytes maturation, but also fertilization, in different models of oocyte-cumulus co-culture. The experimental design is well-conceived,the operation & manipulation is proficient and creditable, the data collection & analysis is standard and precise, and the conclusion is pertinent. So I think the manuscript is acceptable after the following minor revision.
1. Abstract. It has to be reorganized to highlight what’s new and significant. For example, “although no significance is found in xxxxxx, we found xxxxx greatly improve xxxxx. This study suggest that xxxx is important for xxxx”. It’s hard for reader to follow in the current organization.
2. Abbreviation. Cumulus-oocyte complex is usually simplified as COC.
3. Page 4, line 55-56, “Numerous studies have shown that cumulus cells can improve cytoplasmic maturation” , here one or several good review paper should be cited.
4. Page 14, line 312-314, The present study didn’t provide any data about polyspermy, so although polyspermy is an important issue, it shouldn’t be discussed here.

·

Basic reporting

No Comments

Experimental design

No Comments

Validity of the findings

No Comments

Additional comments

The authors aimed to investigate the functions of cumulus cells systematically by different models, employing oocytes with tight junction or dispersed, or without cumulus cells in oocyte maturation and insemination. Their results will improve the outcome in human IVF clinical procedures and research. This is a solid manuscript that should be acceptable.

·

Basic reporting

Overall, it is okay but please see my comments under the section “General Comments for the Author”.

Experimental design

Meets standards but please see my comments under the section “General Comments for the Author”.

Validity of the findings

Under the Statistical Analysis section, the authors may state that experiments were repeated three times.

Additional comments

This study reports that cumulus cells improve both maturation and fertilization of mouse oocytes. The mechanism of oocyte maturation is not fully understood, and in vitro maturation of human oocytes is still considered to be experimental. Therefore, this study is of importance. The following needs attention:
1. The manuscript may benefit from careful editing in terms of readability. For instance, the subtitles might be more descriptive (e.g., Fertilization or Embryonic Development Based on Number of GV or MII Oocytes instead of Total GV or MII Oocytes)
2. It would be helpful to give exact supplementation of culture media used for in vitro maturation and embryo culture (e.g., any serum or albumin supplementation, etc.).
3. The specifics about co-culture are missing (e.g., how many cumulus cells per oocyte; status of cumulus cells in terms expansion and compaction or luteinization, timing of cumulus cell addition, etc.).

Reviewer 5 ·

Basic reporting

Please see below (general comments for the authors)

Experimental design

Please see below (general comments for the authors)

Validity of the findings

Please see below (general comments for the authors)

Additional comments

The study by Zhou et al addresses the influence of cumulus cells on oocyte maturation, fertilization and subsequent development using four different experimental groups: In vivo oocyte-cumulus complexes (OCCs), in vitro OCCs, in vitro denuded oocytes (DOs) plus cumulus cells and in vitro DOs. The authors report that tight connections between cumulus cells and oocytes in OCC improved oocyte nuclear maturation and that the presence of cumulus cells was beneficial for fertilization. They further report that disruption of cell junctions between oocytes and cumulus did not affect fertilization of OCCs matured in vitro. The authors state that cumulus cells play key roles during oocyte maturation, insemination, and early embryo development and that the connections between oocytes and cumulus cells play a positive role during maturation and insemination.

This paper builds on a number of previous reports that established important interactions between oocytes and cumulus cells through gap junctions. As written, the paper requires several improvements and clarifications including the following.

• The title is confusing and needs better wording to more clearly reflect the content of the paper. The title also should indicate that the study was performed in the mouse model.
• In the abstract, the term “connections” should be explained in more scientific and cell biological terms. The sentence “The existence of cumulus promoted matured oocyte fertilization” should include more detail on how fertilization was “promoted” by cumulus cells. Similarly, the sentence “For oocytes matured in vitro, the existence of cumulus cells during maturation or insemination did not contribute to the final blastocyst formation” needs clarification to put this finding in context for the reader to understand the rationale for possible effects on blastocyst formation.
• In the introduction, for the reader to better understand the rationale for the study, it would be important to introduce gap junctions and the interactions between oocytes and cumulus cells in more detail. The paragraph on sperm related to cumulus cells can be shortened.
• In the introduction, cytoplasmic and nuclear maturation should be addressed in better details and explained for the reader to better understand the difference and their interdependence. As written, there is little definition for neither cytoplasmic nor nuclear maturation.
• In the introduction, lines 58-59: “promotional effects” should be explained with more clarity.
• In the introduction, lines 60-61, please clarify: “…..by employ oocyte with tight junctional or dispersed, or without cumulus cells in the procedure of oocyte maturation and insemination….” The sentence is difficult to understand.
• In the introduction, lines 62-63: “This study may provide more detailed application guidelines for cumulus cell use in human IVF clinical practice”. Based on the data presented on the mouse this statement is not justified and should be eliminated.
• In the results, lines 125-126: Please provide more clarity on “the quality of MII oocytes (i.e., the progression beyond the MII stage), and the progress of insemination (i.e., the progression after the MII stage)”.
• In the results, lines 131-137: More detail on the disruption of gap junctions is needed.
• In the results, lines 139-144: Please provide numbers for the findings in the text.
• In the results, lines 169-176: Please provide numbers for the findings in the text.
• Generally in the results section it is not clear why the authors used the word “fraction” of total oocytes.
• The discussion is generally too long and could be shortened substantially (to about half of the text) to address the topic concisely and with clarity, particularly to state the rationale for the study, the significance of the study, the new findings compared to already known information, the importance of nuclear and cytoplasmic maturation with detailed information for both, and more detailed information on the importance, structure and functions of gap junctions between oocytes and cumulus cells, and more information on the rationale for co-culturing oocytes with cumulus cells although connections are not made between the two in these experimental approaches. On the other hand, the speculations on factor secretions from cumulus cells affecting oocytes and others (lines 307-325) should be limited and replaced with information known from the literature. What specifically do the authors mean by “This may be because the gap junctions of OCCs from IVM are incomplete or defective, leading to this dysfunction during IVF”?
• As in the introduction, the sentence “The study of the detailed mechanism of cumulus cell function in oocytes and embryos will improve the outcome in human IVF clinical procedures and research” should be deleted.

---

## Round 0.2 · accepted · Accept

All the reviewers' concerns were addressed.

Reviewer 5 ·

Basic reporting

The authors have addressed this reviewer's points of critique

Experimental design

The authors have addressed this reviewer's points of critique

Validity of the findings

The authors have addressed this reviewer's points of critique

Additional comments

The authors have addressed this reviewer's points of critique